# Vitamin D Metabolism and Its Role in Mineral and Bone Disorders in Chronic Kidney Disease in Humans, Dogs and Cats

**DOI:** 10.3390/metabo10120499

**Published:** 2020-12-04

**Authors:** Fernanda C. Chacar, Márcia M. Kogika, Rafael V. A. Zafalon, Marcio A. Brunetto

**Affiliations:** 1Department of Internal Medicine, Federal Institute of Education, Science and Technology of South of Minas Gerais (IFSULDEMINAS), Muzambinho 37890-000, Brazil; fernanda.chacar@muz.ifsuldeminas.edu.br; 2Department of Internal Medicine, School of Veterinary Medicine and Animal Science, University of Sao Paulo, Sao Paulo 05508-270, Brazil; mmkogika@usp.br; 3Pet Nutrology Research Center, Nutrition and Production Department, School of Veterinary Medicine and Animal Science, University of São Paulo, Pirassununga 13635-900, Brazil; rafael.zafalon@usp.br

**Keywords:** calcidiol, calcitriol, calcium, felines, canines, hyperparathyroidism

## Abstract

Some differences regarding Vitamin D metabolism are described in dogs and cats in comparison with humans, which may be explained by an evolutionary drive among these species. Similarly, vitamin D is one of the most important regulators of mineral metabolism in dogs and cats, as well as in humans. Mineral metabolism is intrinsically related to bone metabolism, thus disturbances in vitamin D have been implicated in the development of chronic kidney disease mineral and bone disorders (CKD-MBD) in people, in addition to dogs and cats. Vitamin D deficiency may be associated with Renal Secondary Hyperparathyroidism (RSHPT), which is the most common mineral disorder in later stages of CKD in dogs and cats. Herein, we review the peculiarities of vitamin D metabolism in these species in comparison with humans, and the role of vitamin D disturbances in the development of CKD-MBD among dogs, cats, and people. Comparative studies may offer some evidence to help further research about vitamin D metabolism and bone disorders in CKD.

## 1. Introduction

In people, chronic kidney disease (CKD) is defined as abnormalities of kidney structure or function for three months or more, with implications for health [1,2,3,4]. In dogs and cats, CKD is characterized by structural or functional abnormalities in one or both kidneys for at least three months, and it courses with a progressive and irreversible loss of nephrons [5,6]. Thus, both definitions of CKD are similar in humans, dogs, and cats.

Dogs have been used as experimental models of CKD for many years. The remnant kidney model has contributed substantially to the understanding of the renal function and allowed the evaluation of the efficacy of therapeutic interventions in CKD [7,8,9,10]. Cats have also been used as a model of human CKD [11,12,13,14]. Both geriatric cats and aging humans progress to chronic tubulointerstitial inflammation and fibrosis as common causes of CKD [14,15].

Vitamin D plays an essential role in the regulation of mineral metabolism, alongside the Klotho/ Fibroblast Growth Factor-23-axis, Parathyroid hormone (PTH), and calcitonin. Mineral and bone metabolism are intrinsically related; thus, vitamin D deficiency may lead to CKD-MBD. Vitamin D deficiency is common in dogs and cats along with the progression of CKD, and the disorders of mineral and bone metabolism occur since the early stages in these species, as well as in people [5,6,7,8].

Comparative studies may offer some information useful for further research about vitamin D metabolism and its disturbances. In this review, we aim to compare the vitamin D metabolism and the disorders of the mineral and bone metabolism in CKD associated with vitamin D abnormalities, among humans, dogs, and cats.

## 2. Chronic Kidney Disease in Humans, Dogs and Cats

The prevalence of CKD is rising worldwide. Cardiovascular morbidity and premature mortality, and high economic costs to the health systems, have been reported as worrisome consequences of CKD in people [3,16,17,18,19,20]. In dogs and cats, CKD is a critical concern as well, since it is associated with reduced survival and impairment to the quality of life [21,22,23,24,25,26,27].

CKD is an irreversible and progressive illness that courses with persistent abnormalities in kidney morphology and function. A similar definition of CKD has been established in human and veterinary medicine, probably due to the resemblance of chronic renal disease between people and small domestic animals [28].

Diabetes, cardiovascular disease, hypertension and obesity are associated with CKD in humans [29,30,31,32,33]. Renal dysplasia, renal amyloidosis, primary glomerulopathies, polycystic kidney disease, and glomerulonephritis have been reported as congenital causes of CKD in dogs and cats, with higher prevalence among specific breeds. Nephrotoxic drugs or toxins, infectious diseases, such as pyelonephritis, renal lymphoma and renal carcinoma, have been described as acquired causes of CKD in these animals [34,35].

A common predisposing factor for CKD among many species of mammals, including humans, dogs and cats is aging [36,37,38,39]. The improvement in healthcare has enhanced the lifespan of people, leading to higher prevalence of age-associated diseases. Similarly, an increase in the longevity of dogs and cats has also been observed, due to the advances in veterinary care [38].

Aged kidneys of humans and dogs usually develop glomerulosclerosis, interstitial fibrosis, and tubular atrophy, which may be related to oxidative stress and inflammation, endothelial dysfunction, and impaired angiogenesis [36,40,41]. In cats, interstitial fibrosis is the most predominant pattern, however the concurrence of secondary glomerulosclerosis has also been reported in the geriatric population and in felines with end-stage renal disease [14,15,42,43,44].

Senescence is an important mechanism involved in the pathogenesis of CKD. Klotho, an anti-aging gene, is involved in many pathways of the senescence processes, and its role in CKD has been under investigation in recent decades [39,41,45,46,47,48]. The overexpression of Klotho resulted in longer life span, whereas its suppression in deficiency mice led to aging-like phenotypes mediated by hyperphosphatemia [47,48].

Kidney calcification, renal cellular apoptosis and decline in renal function were also reported in mice, raising the role of Klotho deficiency in the development of CKD [49,50]. Then, CKD is considered to be a state of accelerated aging associated with hyperphosphatemia induced by Klotho deficiency [49].

The Klotho gene encodes a protein with the same name that is present in the cerebrospinal fluid, blood and urine of mice, humans, and it has recently been found in the urine of dogs [49,51,52,53]. A study performed in dogs with spontaneous CKD showed that urinary klotho was negatively correlated with serum levels of urea, creatinine and phosphorus, and a significant reduction in the urinary klotho-to-creatinine ratio was observed in the later stages of CKD [51]. In people with CKD, the expression of Klotho is reduced as well, and Klotho deficiency has been implicated in CKD-MBD [50,54,55]. In cats, to the best of our knowledge, klotho was only assessed in normal kidney tissue [56].

CKD-MBD may occur since early stages of CKD and it is associated with high mortality rates in humans, dogs and cats [57,58,59]. Therapeutic strategies for CKD aim to retard the onset of mineral and bone disorders, thus avoiding the progression of CKD. Nowadays, the supplementation of Vitamin D analogues in patients with CKD has been widely used, in order to prevent or attenuate the development of CKD-MBD [60,61,62,63,64,65,66]. The deficiency of vitamin D has also been reported in dogs and cats [59,67], and its relation with CKD-MBD in these species is discussed as follows.

## 3. Mineral Metabolism

### 3.1. Vitamin D and PTH: The Main Regulators of Calcium Homeostasis

The regulation of calcium homeostasis is complex. Ionized calcium (iCa) itself, the calcium-sensing receptors (CaSRs) and calcitonin play a fundamental role in calcium control, but the major regulators of calcium homeostasis are PTH and 1,25(OH)_2_D_3_ [68,69,70].

PTH exerts the “minute-to-minute” control of blood iCa levels. This hormone is synthesized by the parathyroid glands in response to low iCa concentrations. The molecular mechanism involved in this process is mediated by the CaSR, which works as a sensor of calcium. Thus, the major physiological role of PTH is to enhance the blood calcium levels by (1) intestinal absorption, (2) renal resorption, and (3) bone resorption [71].

PTH indirectly promotes the absorption of calcium by stimulating the renal 1- alpha-hydroxylase conversion of 25(OH)D (calcidiol) to 1,25(OH)_2_D_3_ (calcitriol). 1,25(OH)_2_D_3_ is the active form of vitamin D_3_, which promotes calcium absorption from the intestine. The 1,25(OH)_2_D_3_ binds to the Vitamin D Receptors (VDRs) and stimulates the calcium influx through the TRPV6 channel, in the apical membrane of the enterocytes, followed by the intracellular transport mediated by calbindin (CaBP-D28K). In addition, the complex 1,25(OH)_2_D_3_/VDRs, Vitamin D receptor activators (VDRAs), also promotes the efflux of calcium to the basolateral membrane, through the Ca-ATPase pump [72,73,74]. 

The PTH-mediated renal resorption of calcium also involves the 1,25(OH)_2_D_3_. The active form of the vitamin D_3_ acts in the distal nephron in a similar way, binding to the VDRs and promoting calcium resorption through the transient receptor potential vanilloid channel type 5 (TRPV5), calbindin-D28K (CaBP-D28K), and sodium-calcium exchanger 1 (NCX1) [75].

Another mechanism of PTH-mediated renal calcium resorption along the nephron is related to PTH receptor 1 (PTHR1). Furthermore, in the proximal tubule, PTH increases the expression of NHE3, which is a cotransporter involved in the renal handling of sodium and calcium. In the Thick Ascending Limb (TAL) the role of PTH in calcium resorption is under investigation, but evidence supports a possible influence on transepithelial voltage [75,76].

Finally, PTH also increases the blood levels of calcium by promoting bone resorption. PTH modulates the osteoclastic activity through the upregulation of the OPG-RANKL-RANK pathway. The Receptor Activator for Nuclear Factor-κB Ligant (RANKL), which is expressed on the surface of stromal cells or osteoblasts, binds to its receptor, the Receptor Activator of Nuclear Factor-κB (RANK), which, in turn, is expressed on the surface of osteoclast progenitors. The link between RANKL and RANK stimulates the differentiation and activation of osteoclasts. On the other hand, osteoprotegerin (OPG) inhibits the osteoclast activation, and thus bone resorption, by acting as a decoy receptor. The decoy receptor works as an inhibitor that blocks the link between the ligand and its regular receptors, thereby avoiding the activation of the receptor complex [77].

PTH upregulates RANKL, whereas it downregulates OPG. The normal function of the VDR complex is critical for PTH-induced osteoclastogenesis and, as well as PTH, 1,25(OH)_2_D_3_ also upregulates the RANKL and inhibits OPG [78] (Figure 1).

In humans, skin exposition to sunlight UV radiation promotes the conversion of 7-dihydrocholesterol (7-DHI) to previtamin D_3_, a precursor of vitamin D metabolism. Interestingly, dogs and cats are not able to perform vitamin D_3_ photosynthesis because they have a higher activity of the enzyme 7-DHI-reductase [79,80,81].

An evolutionary drive has been considered to explain the particular metabolism of vitamin D in those species. Cats have nocturnal behavior and are strict carnivores, whereas dogs are carni-omnivores. They are natural predators, and since prey provides adequate amounts of vitamin D, its endogenous synthesis is not needed [82,83]. Thus, all the body content of vitamin D in dogs and cats is exclusively obtained from diet sources [79,80,81].

There are two dietary sources of vitamin D: ergocalciferol (Vitamin D_2_) and cholecalciferol (Vitamin D_3_). Ergocalciferol is mainly obtained from sun-exposed mushrooms, but small amounts can also be found in plants contaminated with fungi, and it was proven that *Medicago sativa* (alfafa) contains ergocalciferol too [84,85,86,87]. Cholecalciferol is obtained from animal food sources [59,85]. Humans and dogs are able to make use of both ergocalciferol and cholecalciferol, but cats may not [59,88]. Cats discriminate between cholecalciferol and ergocalciferol and have a higher plasmatic concentration of 25(OH)D_3_ than 25(OH)D_2_, which may be attributed to the higher affinity of the vitamin D binding-protein of cats for vitamin D3 metabolites than those of ergocalciferol [88].

After ingestion, cholecalciferol and ergocalciferol are incorporated into chylomicrons and carried to the liver, through the portal system and intestinal lymphatics. In the liver, the enzyme 25-hydroxylase (CYP27A1) converts the cholecalciferol and ergocalciferol to 25(OH)D (calcidiol). 25(OH)D measurement is used to assess the body status of vitamin D, since it is the sum of vitamin D obtained from sun exposure and dietary sources [70,73,77,89].

Recently, it was reported that cats may utilize an epimeric pathway of 25(OH)D_3_. A new metabolite of vitamin D, 3-epi-25(OH)D_3_ was detected in great amounts in the blood circulation of cats, but not in dogs. Then, the measurement of 3-epi-25(OH)D_3_ should be considered in the reliable assessment of vitamin D status in cats [90].

In humans, the half-life of 25(OH)D may vary from two to three weeks [8,23]. An experimental study suggested that the half-life of 25(OH)D may be longer in cats than in humans because of a lower rate of degradation of 25(OH)D [88].

25(OH)D is then uptaken by the Vitamin D-binding protein (VDBP), which transports the 25(OH)D through blood circulation to the kidneys. In the kidneys, the complex 25(OH)D-VDBP is recognized by the megalin–cubilin receptors on the brush borders of the proximal tubules, in an endocytosis-mediated process. After internalization, VDBP is degraded, thus releasing 25(OH)D; 25(OH)D is then converted to 1,25(OH)_2_D_3_ by 1-alpha-hydroxylase (CYP27B1) in the mitochondria [70,73,77].

1,25(OH)_2_D_3_ (calcitriol) is the active form of vitamin D_3_. 1,25(OH)_2_D_3_ then is transported by VDBP-1 to the target tissues, such as the intestine, bone, and kidney itself. The effects of 1,25(OH)_2_D_3_ in the target tissues are mediated by VDRs [70,73,77]. The VDRs are members of the superfamily of nuclear receptors for steroid hormones and act as ligand-activated transcription factors [91,92]. Recently, a prospective blinded study performed in dogs found that these animals have a higher expression of VDRs in the kidney, duodenum and ileum [92].

1,25(OH)_2_D_3_-VDRs promotes calcium absorption from the intestine and kidneys, and calcium resorption from bone, thereby increasing the blood calcium levels. Thus, vitamin D metabolism is intrinsically related to calcium regulation [93,94].

Calcitriol also stimulates NaPi-IIb cotransporter expression, thereby enhancing the intestinal absorption of phosphate, thus increasing the phosphate blood levels; inversely, phosphate inhibits calcitriol synthesis [95].

### 3.2. Diet and Hormonal Control in Phosphorus Metabolism

The most important regulators of phosphorus metabolism are diet, PTH, calcitriol, and the Klotho/FGF-23 axis. More than 70% of phosphorus from food is passively absorbed in the small intestine, especially in the jejunum. The absorption of intestinal phosphorus through active transport also occurs through two mechanisms [96].

The first mechanism involves the Na/K-ATPase pump in the basolateral membrane of the enterocyte, which maintains a favorable electrochemical gradient for phosphorus absorption; the second mechanism involves the type II-b sodium-phosphorus cotransporter (NaPi-IIb) in the apical membrane of the enterocyte that is directly stimulated by calcitriol [70].

The kidneys also have sodium–phosphorus cotransporters. NaPi-IIa and NaPi-IIc are on the brush border of proximal tubules and are stimulated by the hormones and dietary contents of phosphorus. PTH, FGF-23, and the content of phosphorus from the diet inhibit the expression of NaPi-IIa. The dietary contents of phosphorus, magnesium, and FGF-23 inhibit the expression of NaPi-IIc, thus inducing phosphaturia [97].

FGF-23 is a phosphatonin synthesized mainly by osteocytes, whose action depends on its cofactor Klotho. Klotho is a transmembrane protein highly expressed in the kidneys, in which extracellular domain alpha-klotho interacts directly with the receptor of FGF-23, FGFR1c [98]. Osteocytes synthesize FGF-23 in response to increased concentrations of phosphorus in the intestinal lumen, and FGF-23 binds to the FGFR1c in order to form the FGFR-Klotho complex in the distal tubule. Once the complex has been activated, NaPi-IIa and NaPi-IIc expression is then inhibited in the proximal tubule, probably due to paracrine effects [54].

Calciprotein particles (CPPs) seems to be involved in the communication between bone and intestine. The hypothesis is that CPPs could sensitize osteocytes, which would respond with FGF-23 synthesis. The role of CPPs in phosphorus homeostasis remains under investigation, but it is considered that CPPs circulate in blood linked to fetuin A, preventing the precipitation of crystals of calcium phosphate in the extracellular space, especially in the postprandial period [98]. Thus, phosphorus metabolism is regulated by the complex interaction between the intestine, bones and kidneys.

## 4. Mineral and Bone Disorders in Chronic Kidney Disease in Humans, Dogs and Cats—The Role of Vitamin D

Kidneys play a central role in the regulation of mineral metabolism, since they control the urinary excretion of calcium and phosphorus, synthesize calcitriol and are the target organs for PTH and FGF-23. The tissue with the highest expression of Klotho is the distal tubule [55,97].

An elegant study performed by Dr. Kuro-o and his colleagues showed that Klotho knockout mice had an aging phenotype, with atrophied gonads, vascular calcification, and osteoporosis [48]. Later, it has been proved that human patients in advanced stages of CKD had lower urinary immunodetection of Klotho, which was directly correlated to the glomerular filtration rate of those individuals. At that time, the authors concluded that CKD is a permanent state of Klotho deficiency, and the deficiency of Klotho would be the trigger for the development of bone-mineral disorders in CKD [49].

According to the Klothocentric hypothesis, the expression of Klotho decreases as CKD progresses. The continuous loss of renal mass in combination with the lower expression of Klotho, a cofactor required by FGF-23 for binding FGF-23 receptors, induces resistance to the action of FGF-23, thus decreasing phosphaturia and leading to phosphate retention. In order to maintain normophosphatemia, osteocytes continuously synthesize FGF-23. High blood levels of FGF-23 may impair the metabolism of vitamin D, since it inhibits the enzyme 1-alfa-hydroxylase, thus blocking calcitriol synthesis; at the same time, FGF-23 promotes the degradation of calcitriol by 24-hydroxylase stimulation [99,100] (Figure 2).

Calcitriol binds to VDRs in the parathyroid glands and decreases the synthesis and secretion of PTH, which in turn, stimulates the secretion of FGF-23. Some mechanisms involved in the feedback between PTH and FGF-23 have been under investigation. One study showed that the PTH-mediated inhibition of sclerostin via PTH receptors (PTHr) in bone induced the upregulation of Wnt pathway, thus signaling to increase FGF-23 synthesis [101]. Another study found that PTH activates the nuclear receptor-associated protein-1 (Nurr1) in bone cells, leading to an increase in the transcription of FGF-23 [102].

On the other hand, FGF-23 downregulates the synthesis of PTH. FGF-23 binds to the complex Klotho–FGFR1c in the parathyroid glands, thus decreasing the expression of the PTH gene [103]. However, as CKD progresses, the complex Klotho–FGFR1c decreases in the hyperplastic parathyroid glands of uremic individuals, and there is an impairment of the feedback loop between PTH and FGF-23 [104]. Despite the high blood levels of these phosphaturic hormones, a reduction in urinary excretion of phosphate occurs due to the great loss of renal mass in advanced stages of CKD, then leading to phosphate retention and hyperphosphatemia [49,105].

Low circulant levels of 1,25(OH)_2_D_3_, reduced blood concentration of ionized calcium, increased FGF-23 and PTH, phosphate retention and hyperphosphatemia are the main factors involved in the the development of RSHPT [106].

Calcitriol deficiency plays a central role in the development of RSHPT, since it is the major negative regulator of parathyroid function. Calcitriol directly controls the synthesis and secretion of PTH by acting in VDRs in parathyroid glands and indirectly by enhancing intestinal calcium absorption via VDRAs, which, in turn, increases the insertion of the transient receptor potential vanilloid 6 (TRPV6) calcium channel in the apical membrane of enterocytes. Sufficient amounts of calcitriol and iCa inside the parathyroid gland are required to inhibit PTH gene transcription [107].

Like humans, dogs and cats may have vitamin D deficiency since early stages of CKD, due to decreased nutritional intake, increased inflammatory cytokines, and increased FGF-23 [57,59,67,108].

A study showed that dogs with naturally occurring CKD had lower serum concentrations of 1,25(OH)_2_D, 25(OH)D, and 24,25(OH)_2_D than healthy controls, however, differences were noted only in later stages of CKD. Dogs with CKD had higher values of FGF-23, which were negatively correlated with vitamin D metabolites. Despite high serum concentrations of FGF-23, low levels of 24,25(OH)_2_D were found. FGF-23 stimulates 24-hydroxylase, thereby enhancing the conversion of 25(OH)D to 24,25(OH)_2_D, then higher values of 24,25(OH)_2_D were expected [59].

Another study found that dogs with mild and moderate CKD had increased values of FGF-23 in comparison with healthy controls. A high 25(OH)D_3_ to 24,25(OH)_2_D_3_ ratio was also observed in dogs with mild and moderate CKD, thus failing to confirm the FGF23-mediated catabolism of vitamin D metabolites [109].

In CKD cats with nephrolithiasis, a report found a high 25(OH)D_3_ to 24,25(OH)_2_D_3_ ratio, but FGF-23 was not assessed [110]. Studies have shown that FGF-23 is increased since the early stages of CKD in cats; thus, FGF-23 was expected to be increased in those cases [58,111,112].

It is important to note that none of the studies above used a standardized diet, which would be of great value considering that all the body content of vitamin D in dogs and cats is exclusively obtained from dietary sources. Nutritional studies with a proper design are needed to investigate whether the content of vitamin D in commercial pet foods may be associated with changes in calcitriol levels in dogs and cats with CKD.

In an open-label prospective study, people with CKD had lower serum concentrations of 24,25(OH)_2_D_3_ and an equivalent rise in the 25(OH)D_3_ substrate, in response to an equal dose of cholecalciferol, in comparison with healthy controls. These findings suggested that the regulation of 24-hydroxylase may be altered in patients with poor kidney function [113].

The reduced renal function observed in later stages of CKD leads to uremia that is strongly related to changes in vitamin D metabolism, which, in turn, is associated with mineral and bone disorders. Uremic toxins may induce resistance to the action of 1,25(OH)_2_D_3_ by decreasing the binding sites for the VDR to the osteocalcin VDRE [114]. Moreover, uremia may impair the hepatic synthesis of calcidiol, due to the PTH-mediated inhibition of CYP450 isoforms [115,116].

The mineral and bone metabolism are intrinsically related, thus mineral and bone disorders are common in patients with CKD. CKD-MBD is a syndrome characterized by clinical, laboratorial (e.g., changes in blood levels of calcium, PTH, Vitamin D, phosphate and FGF-23), bone and vascular abnormalities. CKD-MBD is associated with poor quality of life and increased morbidity and mortality [106].

RSHPT is the most common mineral disorders in dogs and cats. The overall prevalence of RSHPT in these species varied from 76% to 84%, and it was reported 100% of RSHPT in dogs and cats with end-stage CKD [67,106,117,118].

PTH and 1,25(OH)_2_D_3_ are prime regulators of bone remodeling. The process of coupling between bone formation and bone resorption is disrupted in RSHPT, resulting in resorptive lesions and low bone mineral density, which, in turn, lead to an increase in bone fragility [118,119]. Dogs usually develop renal osteodystrophy in association with RSHPT. However, in cats, bone changes in CKD are rarely reported [120]. In humans, it is well-established that vitamin D supplementation or its synthetic analogues in earlier stages of CKD avoid the development of mineral and bone disorders and increase survival time [121,122].

In dogs, a double-masked, randomized, controlled clinical trial was performed in thirty-seven dogs with spontaneous CKD with stages 3 and 4. It was found that calcitriol therapy was associated with a significant reduction in all-cause mortality, and the median survival time was 365 days for dogs receiving calcitriol, whereas it was 250 days for dogs receiving placebo [123]. In cats with CKD, daily and intermittent-dose calcitriol did not result in significant differences in serum PTH before and after treatment [124]. Thus, more evidence is needed to support the use of vitamin D in earlier stages of CKD in dogs and cats, in order to prevent CKD-MBD.

We summarized the mainly peculiarities of vitamin D metabolism and the role of vitamin D disturbances in the development of CKD-MBD among dogs, cats, and people in the figure below (Figure 3).

## 5. Conclusions

Humans can synthesize vitamin D from sun exposure, whereas dogs and cats are not able to. Thus, all the body content of vitamin D in these animals is obtained exclusively from food, which emphasizes the role of diet in the evaluation of vitamin D status in these cases; in addition, the measurement of 3-epi-25(OH)D_3_ should be considered in cats.

The effect of FGF-23 in the catabolism of vitamin D metabolites seems similar in humans and dogs since different studies performed in both species have failed to show the FGF-23-mediated stimulus of 24-hydroxylase activity, through the 25(OH)D_3_ to 24,25(OH)_2_D_3_ ratio. Anecdotal data are also available for cats, but more evidence is needed.

As in humans, vitamin D disturbances have been implicated in the development of CKD-MBD in dogs and cats. The main disorder in these animals is the RSHPT and, as in humans, it usually occurs in later stages of CKD. The most common bone disorder of that syndrome in dogs is renal osteodystrophy, which is rarely reported in cats.

In conclusion, humans, dogs, and cats have similarities in vitamin D metabolism. Differences might be related to an evolutionary driven among these species. Comparative studies may be useful to elucidate the vitamin D metabolism and its role in CKD-MBD.

## Figures and Tables

**Figure 1 metabolites-10-00499-f001:**
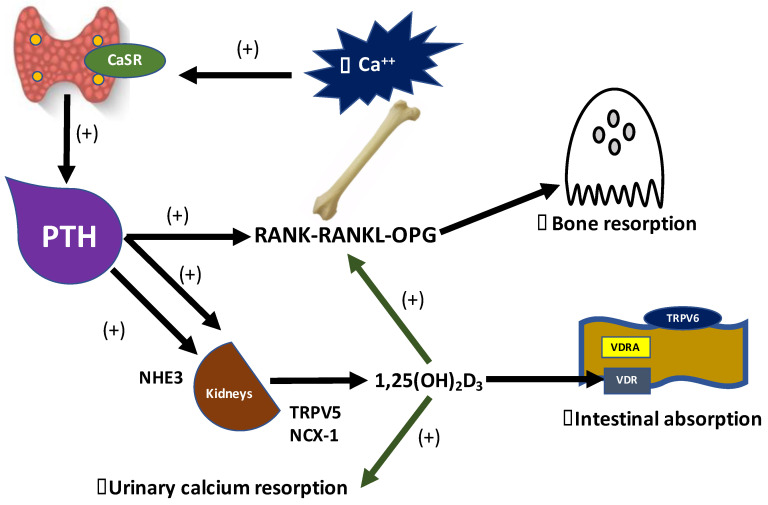
Vitamin D and Parathyroid hormone (PTH) PTH-mediated calcium absorption in intestine, kidneys and bones. CaSRs in the parathyroid glands detect lower levels of blood ionized calcium, leading to the synthesis and secretion of PTH, which in turn promotes the conversion of 25(OH)D to 1,25(OH)_2_D_3_ by stimulating 1-alpha-hydroxylase in proximal tubules. 1,25(OH)_2_D_3_ is the active form of vitamin D_3_ that acts increasing the expression of TRPV6 in the apical membrane of enterocytes, via 1,25(OH)_2_D_3_-VDR complex, thus leading to the intestinal absorption of calcium. The 1,25(OH)_2_D_3_-VDR complex increases the urinary calcium resorption from distal nephron through the luminal insertion of TRPV5, and NCX-1 in the basolateral cellular membrane. PTH directly promotes the urinary calcium resorption by stimulates NHE3 in proximal tubules. PTH and 1,25(OH)_2_D_3_ upregulate the RANKL and inhibit OPG, then increasing calcium resorption from bones.

**Figure 2 metabolites-10-00499-f002:**
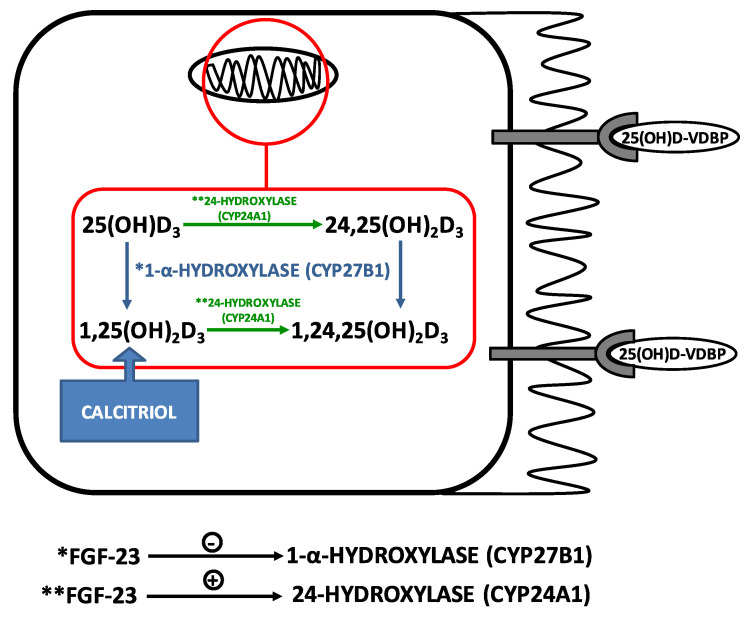
Regulation of Vitamin D metabolism by FGF-23. The megabin-cubilin receptors on the brush border of proximal tubules uptake the complex 25(OH)D-VDBP in an endocytosis-mediated process. In mitochondria, 25(OH)D_3_ is converted to the active form 1,25(OH)_2_D_3_ by 1-alpha-hydroxylase. FGF-23 inhibits 1-alpha-hydroxylase and stimulates 24-hydroxylase, which degrades both 25(OH)D_3_ and 1,25(OH)_2_D_3_ to 24,25(OH)_2_D_3_ and 1,24,25(OH)_2_D_3_, respectively. Note: *action of 1-alpha-hydroxylase (CYP27B1); ** action of 24-hydroxylase (CYP24A1).

**Figure 3 metabolites-10-00499-f003:**
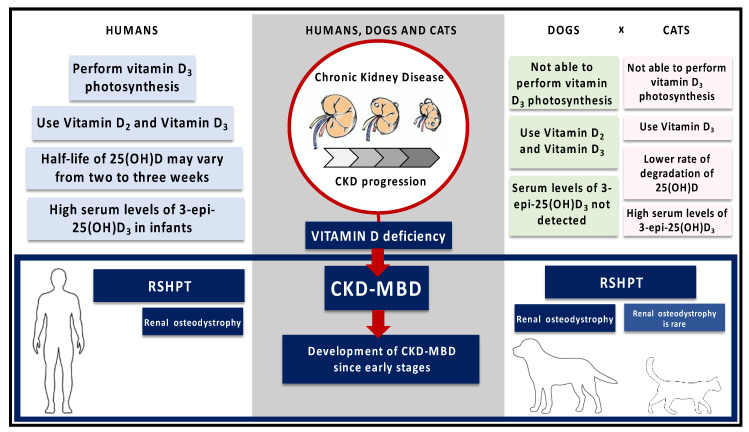
Comparison of vitamin D metabolism and the role of vitamin D disturbances in the development of CKD-MBD among dogs, cats, and people. Humans are able to perform vitamin D_3_ photosynthesis, whereas dogs and cats are not [79,80,81]. Both dogs and humans use Vitamin D_2_ and Vitamin D_3_, but cats discriminate between these two metabolites [88]. In comparison with humans, cats may have a longer half-life of 25(OH)D because of a lower rate of degradation of 25(OH)D [88]. High serum levels of 3-epi-25(OH)D_3_ were previously reported in infants, and it has recently been found great amounts of 3-epi-25(OH)D_3_ in the serum of cats [90,125]. The vitamin D deficiency is associated with the CKD-MBD since early stages in humans, dogs and cats. RSHPT commonly occurs in these species, but the development of renal osteodystrophy is rare in cats [118,120].

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
