# Peer review of "Vitamin D Metabolism and Its Role in Mineral and Bone Disorders in Chronic Kidney Disease in Humans, Dogs and Cats"

_metabolites, 2020, doi:10.3390/metabo10120499_

Round 1

Reviewer 1 Report

The authors present a review of Vitamin D metabolism in CKD and CKD-MBD comparing physiology and disease in humans as well as animals.

The review is well written and organized. It is a sound, significant contribution in the field of Vitamin D metabolism.

There are only minor issues in wording, like line 62 (multiple use of "congenital causes").

I fully recommend to publish this review.

Author Response

Response to Reviewer 1 Comments

Point 1: There are only minor issues in wording, like line 62 (multiple use of "congenital causes").

Response 1: We excluded “congenital causes” at the beginning of the sentence. Please see the attachment.

Reviewer 2 Report

This is potentially a useful review with the novel approach of comparing chronic kidney disease in humans with this condition in dogs and cats. However, there are a number of examples where information is quoted to come from other reviews, and it appears that this has introduced the risk of erroneous interpretations being compounded. This would be a more valuable review if the references quoted could be to the original papers and not to other reviews. There are a number of points that the authors should consider in revising this manuscript:

  1. At line 76 there is reference to Klotho as an antiaging gene, yet at line 84 the term “urinary klotho” is used. If this is the gene product of the Klotho gene then its chemical nature should be defined at this point.
  2. Line 100: “major regulators of calcium metabolism are PTH and vitamin D metabolites” There is only one vitamin D metabolite, and not more than one as implied here, that is a major regulator of calcium transport into and out of cells. The term “metabolism” referring to calcium is also a misuse of this word. Metabolism refers to enzyme catalysed chemical reactions in which chemical bonds are formed or broken. The action of 1,25-dihydroxyvitamin D is on the process of transport of calcium across cell membranes and not on the formation or breaking of chemical bonds.
  3. Line 153: “vitamin D stored in the liver” This is incorrect. There is no store of vitamin D in the liver. Any vitamin D found in that organ is either en route to metabolic inactivation or else it is about to undergo 25-hydroxylation and the 25-hydroxyvitamin D is soon to be released into the circulation. Any reference that claims that liver is a store of vitamin D should provide evidence that this is the case.
  4. Line 157: “Ergocalciferol is obtained from plants”. This is incorrect. Ergocalciferol has only been reported in organisms that contain its precursor, ergosterol. Ergosterol is only found in yeasts and fungi. Yes, ergocalciferol can be produced in mushrooms because these organisms are fungi and contain ergosterol, but green plants do not.
  5. Line 175-176: “In the liver, 25(OH)D forms a complex with the Vitamin D-binding protein (VDBP), which transports the 25(OH)D to the kidneys.” This is not correct. Both the vitamin D-binding protein and 25-hydroxyvitamin D are produced in the liver but they emerge independently and 25-hydroxyvitamin D only binds to the vitamin D-binding protein when in the circulation. There are many more vitamin D-binding protein molecules in blood than 25-hydroxyvitamin D molecules. The concentration of 25-hydroxyvitamin D in blood is less than 5% of the concentration of vitamin D-binding protein.
  6. Lines 176-179: The statement that the complex of 25-hydroxyvitamin D bound to the vitamin D binding protein is the mechanism for delivering 25-hydroxyvitamin D to renal cells is incorrect. As stated in point 5 above, less than 5% of the vitamin D-binding protein in blood has a 25-hydroxyvitamin D molecule bound to it. All the evidence suggests that 25-hydroxyvitamin D enters the renal cells by membrane diffusion, although it is possible that this diffusion is enhanced by the presence of internalised vitamin D-binding protein in the cytoplasm causing a gradient of unbound 25-hydroxyvitamin D between outside and inside the renal cells.
  7. The title of the paper indicates that this is a comparison of chronic kidney disease in humans, dogs and cats. It is very difficult to see the similarities and differences between chronic renal disease in these 3 species. It would be helpful if some summary could be provided which shows the similarities and differences between these species. The impression in the current manuscript is that they are so similar that there is no point in comparing these species.

Author Response

Response to Reviewer 2 Comments

Point 1: At line 76 there is reference to Klotho as an antiaging gene, yet at line 84 the term “urinary klotho” is used. If this is the gene product of the Klotho gene then its chemical nature should be defined at this point.

Response 1: We add the sentence “The Klotho gene encodes a protein with the same name that is present in cerebrospinal fluid, blood and urine of mice, humans, and recently, it has been found in the urine of dogs”. We discuss the chemical nature of Klotho protein in the lines 211-212.

Point 2: Line 100: “major regulators of calcium metabolism are PTH and vitamin D metabolites” There is only one vitamin D metabolite, and not more than one as implied here, that is a major regulator of calcium transport into and out of cells. The term “metabolism” referring to calcium is also a misuse of this word. Metabolism refers to enzyme catalyzed chemical reactions in which chemical bonds are formed or broken. The action of 1,25-dihydroxyvitamin D is on the process of transport of calcium across cell membranes and not on the formation or breaking of chemical bonds.

Response 2: We replaced “Vitamin D metabolites” with 1,25(OH)2D3. In fact, the only metabolite that directly regulates the calcium homeostasis is calcitriol, which is the active form of vitamin D; We replaced “calcium metabolism” with “calcium homeostasis”.

Point 3: Line 153: “vitamin D stored in the liver” This is incorrect. There is no store of vitamin D in the liver. Any vitamin D found in that organ is either en route to metabolic inactivation or else it is about to undergo 25-hydroxylation and the 25-hydroxyvitamin D is soon to be released into the circulation. Any reference that claims that liver is a store of vitamin D should provide evidence that this is the case.

Response 3: We rewrote the sentence based on the manuscript of Morris (2002) that says “Analyses of the vitamin D concentration of potential prey of cats (rodents and birds) indicated that the prey could provide adequate amounts of the vitamin without the need for endogenous synthesis”. (Please, see the reference 83).  

Point 4: Line 157: “Ergocalciferol is obtained from plants”. This is incorrect. Ergocalciferol has only been reported in organisms that contain its precursor, ergosterol. Ergosterol is only found in yeasts and fungi. Yes, ergocalciferol can be produced in mushrooms because these organisms are fungi and contain ergosterol, but green plants do not.

Response 4: Oonincx and co-authors (2018) say in their manuscript that "Ergosterol is the primary vitamin D precursor in plants, yeasts and fungi, which UVb light converts to vitamin D2” (please, see the reference 87). It was reported by Jäpelt and Jette Jakobsen(2013) that "small amounts of ergocalciferol can be found in plants contaminated with fungi and traditionally only Vitamin D2 has been considered present in plants" (please, see the reference 86). Horst and colleagues (1984) proved in an elegant study that Vitamin D2 was found in alfafa, which is a leguminous (please, see the reference 84).

Point 5: “In the liver, 25(OH)D forms a complex with the Vitamin D-binding protein (VDBP), which transports the 25(OH)D to the kidneys.” This is not correct. Both the vitamin D-binding protein and 25-hydroxyvitamin D are produced in the liver but they emerge independently and 25-hydroxyvitamin D only binds to the vitamin D-binding protein when in the circulation. There are many more vitamin D-binding protein molecules in blood than 25-hydroxyvitamin D molecules. The concentration of 25-hydroxyvitamin D in blood is less than 5% of the concentration of vitamin D-binding protein.

Response 5: Thank you for the comment, we rewrote the sentence: “25(OH)D is then uptaken by the Vitamin D-binding protein (VDBP), which transports the 25(OH)D throught blood circulation to the kidneys”.

Point 6: Lines 176-179: The statement that the complex of 25-hydroxyvitamin D bound to the vitamin D binding protein is the mechanism for delivering 25-hydroxyvitamin D to renal cells is incorrect. As stated in point 5 above, less than 5% of the vitamin D-binding protein in blood has a 25-hydroxyvitamin D molecule bound to it. All the evidence suggests that 25-hydroxyvitamin D enters the renal cells by membrane diffusion, although it is possible that this diffusion is enhanced by the presence of internalised vitamin D-binding protein in the cytoplasm causing a gradient of unbound 25-hydroxyvitamin D between outside and inside the renal cells.

Response 6: In accordance with Bikle and Schwartz (2019), “(…) Nearly all 25(OH)D circulates as the bound form, with the vitamin D binding protein (DBP) accounting for approximately 85% of the binding, with albumin accounting for most of the rest”(…) In a normal non-pregnant individual, approximately 0.03% of 25(OH)D is free; 85% is bound to DBP, 15% is bound to albumin”. These authors also say that “based on the free hormone hypothesis, 25(OH)D can enter cells of most tissues (…) However, tissues expressing the megalin/cubilin complex, such as the kidney, have the capability of taking up 25(OH)D still bound to DBP” (Bikle DD, Schwartz J. Vitamin D Binding Protein, Total and Free Vitamin D Levels in Different Physiological and Pathophysiological Conditions. Front Endocrinol. 2019;10:317).

In another study performed with VDBP null mice, the authors concluded that “DBP’s primary role appeared to be the sequestration of vitamin D sterols in the serum, prolonging their serum half-lives and providing a circulating store of 25(OH)D to meet transient periods of vitamin D deficiency. In so doing, DBP helped to prevent the development of severe vitamin D deficiency that leads to bone mineralization defects and the ultimate development of osteomalacia or rickets. As an adjunct to its role in conservation and optimization of vitamin D, DBP appeared to minimize direct urinary losses of the sterols and to slow their entry into metabolic breakdown pathways. One corollary of this phenomenon is that DBP -/- mice, totally lacking DBP, manifested a need for continual intake of vitamin D to avoid the deficiency syndrome and also a reciprocal resistance to vitamin D toxicity in the face of increased sterol load”. (Safadi FF, Thornton P, Magiera H, Hollis BW, Gentile M, Haddad JG, et al. Osteopathy and resistance to vitamin D toxicity in mice null for vitamin D binding protein. J Clin Invest. 1999;103(2):239–51).

Point 7: The title of the paper indicates that this is a comparison of chronic kidney disease in humans, dogs and cats. It is very difficult to see the similarities and differences between chronic renal disease in these 3 species. It would be helpful if some summary could be provided which shows the similarities and differences between these species. The impression in the current manuscript is that they are so similar that there is no point in comparing these species.

Response 7: The title of our paper is "Vitamin D Metabolism and Its Role in Mineral and Bone Disorders in Chronic Kidney Disease: A Comparative Approach Among Humans, Dogs and Cats". Our aim is to review the peculiarities of vitamin D metabolism in these species in comparison with humans, and the role of vitamin D disturbances in the development of CKD-MBD among dogs, cats, and people. We provided a graphical abstract showing the similarities and differences between these species (please, see the line 336).

Round 2

Reviewer 2 Report

The title of the paper suggests that there are significant differences in aspects of chronic renal disease between the 3 species. However the Figure 3 diagram (lines 336-349) really indicates that the differences are trivial. A more accurate title would therefore be

"Vitamin D Metabolism and Its Role in Mineral and Bone Disorders in Chronic Kidney Disease in Humans, Dogs and Cats"

Although many papers are quoted in support of the topics being raised in the extensive narrative, these in some cases are not critically assessed. An example of this is the assertion that vitamin D2 can be obtained by eating green plants, a conclusion that can certainly be challenged.

Author Response

Response to Reviewer 2 Comments (Round 2):

Comments and Suggestions for Authors: The title of the paper suggests that there are significant differences in aspects of chronic renal disease between the 3 species. However the Figure 3 diagram (lines 336-349) really indicates that the differences are trivial. A more accurate title would therefore be

"Vitamin D Metabolism and Its Role in Mineral and Bone Disorders in Chronic Kidney Disease in Humans, Dogs and Cats"

Response: Thank you for the comment. We have changed the title as you suggested.

This manuscript is a resubmission of an earlier submission. The following is a list of the peer review reports and author responses from that submission.